# *KOMPEITO*, an Atypical *Arabidopsis* Rhomboid-Related Gene, Is Required for Callose Accumulation and Pollen Wall Development

**DOI:** 10.3390/ijms23115959

**Published:** 2022-05-25

**Authors:** Masahiro M. Kanaoka, Kentaro K. Shimizu, Bo Xie, Sinisa Urban, Matthew Freeman, Zonglie Hong, Kiyotaka Okada

**Affiliations:** 1Department of Botany, Graduate School of Science, Kyoto University, Kitashirakawa-Oiwake-Cho, Sakyo-Ku, Kyoto 606-8502, Japan; kentaro.shimizu@uzh.ch (K.K.S.); kiyo@ad.ryukoku.ac.jp (K.O.); 2Department of Microbiology, Molecular Biology and Biochemistry, University of Idaho, Moscow, ID 83844, USA; xiebo@mail.ccnu.edu.cn (B.X.); zhongl@uidaho.edu (Z.H.); 3Division of Biological Science, Graduate School of Science, Nagoya University, Furo-cho, Chikusa-ku, Nagoya 464-8602, Japan; 4Department of Evolutionary Biology and Environmental Studies, University of Zurich, Winterthurerstrasse 190, CH-8057 Zurich, Switzerland; 5Kihara Institute for Biological Research, Yokohama City University, Yokohama 244-0813, Japan; 6Department of Molecular Biology and Genetics, Johns Hopkins University School of Medicine, Baltimore, MD 21205, USA; surban@jhmi.edu; 7Dunn School of Pathology, University of Oxford, South Parks Road, Oxford OX1 3RE, UK; matthew.freeman@path.ox.ac.uk; 8Ryukoku Extension Center (REC) Ryukoku University, Yokotani 1-5, Seta Ohe-cho, Otsu-shi 520-2194, Japan; 9Core Research of Science and Technology (CREST) Research Project, Tokyo 102-0076, Japan

**Keywords:** *Arabidopsis*, callose, meiosis, pollen development, exine, Rhomboid

## Abstract

Fertilization is a key event for sexually reproducing plants. Pollen–stigma adhesion, which is the first step in male–female interaction during fertilization, requires proper pollen wall patterning. Callose, which is a β-1.3-glucan, is an essential polysaccharide that is required for pollen development and pollen wall formation. Mutations in *CALLOSE SYNTHASE 5* (*CalS5*) disrupt male meiotic callose accumulation; however, how CalS5 activity and callose synthesis are regulated is not fully understood. In this paper, we report the isolation of a *kompeito-1* (*kom-1*) mutant defective in pollen wall patterning and pollen–stigma adhesion in *Arabidopsis thaliana*. Callose was not accumulated in *kom-1* meiocytes or microspores, which was very similar to the *cals5* mutant. The *KOM* gene encoded a member of a subclass of Rhomboid serine protease proteins that lacked active site residues. KOM was localized to the Golgi apparatus, and both *KOM* and *CalS5* genes were highly expressed in meiocytes. A 220 kDa CalS5 protein was detected in wild-type (Col-0) floral buds but was dramatically reduced in *kom-1*. These results suggested that KOM was required for CalS5 protein accumulation, leading to the regulation of meiocyte-specific callose accumulation and pollen wall formation.

## 1. Introduction

In higher plants, coordinated interactions between male and female reproductive organs are crucial for fertilization. As the first step of fertilization, pollen grains formed in male anthers are conveyed and attached to the stigmatic papillae cells of female pistils. This pollen–stigma adhesion event is tightly regulated, promoting crossing within species and preventing interspecies crossing. One of the key components in this process is exine, which is the surface coating of pollen grains; purified exine was shown to retain the ability to bind to the stigma [1]. 

Several genes and factors were reported to affect pollen wall formation and male fertility in many plant species [2,3,4,5,6,7,8]. Of these, callose plays an important role in pollen wall formation [2,6,8,9,10]. Callose, a β-1,3-glucan, is synthesized by callose synthases (CalS) [10,11,12,13,14]. Of the 12 *CalS* genes in *Arabidopsis*, *CalS5* is responsible for the synthesis of callose during pollen development, especially during the process of pollen wall formation [2,15,16,17,18]. The development of pollen mother cells (meiocytes) requires the synthesis of a large amount of callose, which is deposited between the primary cell wall and plasma membrane, forming a new, specialized cell wall known as the callose wall. The callose wall becomes thickened after meiosis and encloses the tetrad microspores; the loss of this wall leads to the formation of microspores with multiple nuclei [12,19,20,21,22]. The callose wall also functions to prevent the adhesion of microspores to the locule during pollen development [23,24]. During microsporogenesis, the callose wall is degraded by tapetum-derived β-1,3-glucanases (callase), resulting in the release of microspores from the tetrads to the locular space. The overexpression of callase in tapetum cells was found to reduce callose deposition during microsporogenesis, causing a disruption in pollen wall formation [23,24]. Thus, the timing of callose deposition and removal must be tightly regulated during pollen formation. 

Multiple lines of evidence suggest that callose synthesis is spatiotemporally regulated. In cotton, the SKS-like protein PSP231 triggers and fine-tunes callose synthesis and deposition [25]. In peas, a 145 kDa protein, namely, CFL1, has callose synthase activity and its molecular weight is lower than that predicted from the gene sequence [26,27]. Partial treatment of callose synthase with trypsin from tobacco pollen tubes in the presence of digitonin was shown to cause the activation of CalS activity. CDKG1, which encodes a cyclin-dependent kinase, is associated with the spliceosome and regulates CalS5 pre-mRNA splicing and protein maturation [28]. However, the mechanism by which callose synthase activity and callose production are regulated during microsporogenesis is not fully understood. 

We hypothesized that as yet unknown factor(s) contribute to regulating pollen and callose wall development. To identify such factor(s), we aimed to find mutants that were defective in this process. In this article, we report a *kompeito-1* (*kom-1*) mutant that was defective in pollen wall patterning and pollen–stigma adhesion in *Arabidopsis thaliana*. We then performed genetic and molecular analyses of *KOMPEITO* (KOM), a novel member of the Rhomboid protein family, to clarify its role in callose accumulation and pollen wall development.

## 2. Results

### 2.1. Isolation of Kompeito-1 (Kom-1), Which Was Defective in Pollen Wall and Pollen–Stigma Adhesion 

We screened mutants that were defective in plant fertility in an ethyl methanesulfonate (EMS)-induced *Arabidopsis* population. In the M2 population of EMS-treated plants, we found plants with shorter siliques (Figure 1A,B) and fewer seeds per silique. Pollen grains of this mutant lacked regular, mesh-like pollen wall patterning and had dot-like sporopollenin deposits on their surfaces (Figure 1E,F), as shown with the Auramine-O staining, which reacts with sporopollenin [29]. This mutant was named *kompeito-1* (*kom-1*) because the shape of the pollen grains resembled a Portuguese candy, namely, *kompeito* or confeito, in their scanning electron microscopy images (Figure 1C,D). Transmission electron microscopy (TEM) analyses of pollen exines revealed that the statue-like structures, i.e., tecta (Figure 1G, arrowheads), were absent from the surface of the *kom-1* pollen grains (Figure 1H). When wild-type (WT) Col-0 pistils were hand-pollinated with WT pollen and washed 10 s after pollination, 71.3% of pollen grains remained on the stigma. In contrast, when WT pistils pollinated with *kom-1* pollen grains were washed 10 s after pollination, only 16.9% of pollen grains remained (Figure 1I). When the same washing experiment was performed 1 h after pollination, which is enough time for pollen tubes to germinate and penetrate inside the stigma, 89.0% of WT and 83.7% of *kom-1* pollen grains remained (Figure 1I). These data suggested that it was difficult for *kom-1* pollen grains to be retained on the stigma surface, but once retained, they could germinate normally. 

The vegetative parts of the *kom-1* mutant were indistinguishable from the WT plants, and its female gametophytes developed normally (Appendix A). Despite the abnormality of the pollen surface, when the *kom-1* pollen grains were pollinated onto the WT pistil, the pistil produced seeds normally. This suggested that the male *kom-1* gametophytes were functional (Appendix A). When the *kom-1* pollen grains or pistils were crossed with the WT plants, all of the F1 progeny had the normal pollen phenotype and no mutant plants were segregated. When the F1 plant was self-pollinated, 24.7% (24 out of 97) of the F2 plants showed a *kom-1* mutant phenotype. These data indicated that *kom-1* was a recessive, sporophytic, but not gametophytic mutant, and that the wall patterning of the pollen grains was important for pollen–stigma adhesion.

### 2.2. Defective Callose Wall Development in Kom-1

We analyzed the composition and structure of the callose wall in developing *kom-1* anthers stained with aniline blue, which is a callose indicator (Figure 2). Pollen mother cells and microspore tetrads of the WT plants were enclosed with a thick wall of callose (Figure 2B,C). No callose was stained in the cell wall surrounding the *kom-1* pollen mother cells or tetrads (Figure 2E); however, a moderate amount of callose was observed in the walls separating the *kom-1* microspores (Figure 2F). TEM analyses of the callose wall ultrastructure further confirmed that the peripheral wall surrounding the *kom-1* tetrad was thin and lacked callose (Figure 2J,K) compared with the thick wall of the WT (Figure 2G,H). TEM images also showed that the walls within and between the microspores were thinner and less electron-dense (Figure 2K). The pattern of sporopollenin aggregation observed at the tetrad stage in the WT (Figure 2I) did not form in *kom-1* (Figure 2L). These results suggested that KOM was responsible for the synthesis of the callose wall surrounding the tetrad during meiosis, but was not absolutely necessary for the callose wall that was synthesized in the tetrad stage and secreted between microspores. 

Since female meiosis also accompanies the formation of callose walls, we wondered whether megasporogenesis was affected by the *kom-1* mutation. The amount of callose in the callose walls separating the megaspore mother cells was reduced in *kom-1* (Appendix A) compared with that of the wildtype, which had thick callose walls (Appendix A). These results suggested that KOM was also important for callose accumulation during megasporogenesis. Surprisingly, female *kom-1* gametophytes developed to become indistinguishable from those of the WT (Appendix A) and were fertile (Appendix A). One possible explanation is that three of the four megaspore mother cells present after meiosis undergo apoptosis during megasporogenesis; thus, callose reduction may not produce defects in female gametogenesis.

### 2.3. Mapping of the Kom-1 Locus and Complementation Using the KOM Genomic DNA

To isolate the gene responsible for the recessive *kom-1* mutation, we mapped the locus using polymerase chain reaction (PCR)-based polymorphism detection methods with DNA samples prepared from F2 plants of a cross between *kom-1* (Col-0 ecotype) and Landsberg *erecta* (L*er*). A single locus at the bottom of chromosome 1 was linked to the *kom-1* phenotype (Figure 3A). After fine mapping, the *kom-1* locus was narrowed down to a 12.5 kb region of BAC F28K19, which contained four open-reading frames. We sequenced the genomic DNA of this region from the *kom-1* plant and found only one G-to-A substitution in the At1g77860 gene. We cloned the full-length cDNAs of At1g77860 from WT plants and the *kom-1* mutant. A comparison of the genomic and cDNA sequences revealed that the mutation site was in the splicing acceptor (AG) of the fourth intron, which disabled normal splicing of the *KOM* gene (Figure 3B). In *kom-1* plants, an alternative splicing site 23 bp downward from the original site was used as a new splice acceptor, resulting in a frameshift that produced a truncated protein with 16 novel amino acids at its C-terminal end. The resulting *kom-1* mRNA encoded a truncated protein that was approximately 70 amino acids shorter at the C terminus (Figure 3C). This observation indicated that the C-terminus of KOM was essential for its biological function. 

We obtained two additional lines of evidence to support the conclusion that At1g77860 encodes the *KOM* gene. First, we performed a complementation test of the kom-1 mutation using a 3.6 kb genomic fragment that spanned the *KOM* gene, including the promoter region and 3’ element. The transgenic plants were restored to WT levels of callose accumulation and pollen wall patterning (Figure 3E–H). Second, a similar phenotype was observed in an allelic mutant, namely, SALK_016980 (*kom-2*), which contained a T-DNA insertion in the fourth exon. The F1 plants of the cross between *kom-1* and *kom-2* did not complement the mutant phenotype (data not shown), suggesting that *kom-1* and *kom-2* were allelic and loss-of-function mutations.

### 2.4. KOM Was Found to Be a Unique Member of the Rhomboid Family of Proteins

The deduced KOM peptide encoded 385 amino acid residues with a signal anchor domain at its N-terminus and 7 predicted transmembrane domains; it belonged to the Rhomboid family of proteins (Figure 3D), which are conserved in bacteria, animals and plants [30,31,32,33,34,35,36,37]. Unlike many members of the Rhomboid family [31,38], KOM did not contain a conserved His residue at TMD6, which, in all other Rhomboids, is required for the serine protease activity. In addition, KOM had a relatively long tail at the C-terminus [31]. Phylogenetic analysis showed that KOM is a distantly-related member of *Arabidopsis* Rhomboid proteins (Appendix A).

### 2.5. Expression of KOM-1 Was Restricted to the Meiocytes

Reverse-transcription PCR analysis showed that *KOM* was mainly expressed in flowers (Figure 4A, lane 6), as well as in the roots and rosette leaves at low levels (Figure 4A, lanes 2 and 3). This expression pattern was also supported by publicly-available RNAseq data (Appendix A). Results of in situ RNA hybridization of floral tissues showed that *KOM* mRNA expression was restricted to pollen mother cells that had undergone meiosis (Figure 4C). *KOM* expression was not detectable in tapetum cells (Figure 4B) or tetrads (Figure 4D). *KOM* mRNA was also present in megaspore mother cells under meiosis (Appendix A). This temporal and spatial expression pattern of the *KOM* gene was consistent with its function in the regulation of callose deposition to the callose wall.

### 2.6. KOM Was Localized in the Golgi Apparatus

Most Rhomboid-like proteins are distributed through the secretory pathway [39], although some members act in mitochondria [33,40]. *Arabidopsis* Rhomboid-like proteins were also experimentally shown or predicted to localize several subcellular organelles [31,41,42]. To investigate the subcellular localization of KOM, we expressed the KOM-GFP fusion protein transiently in *Arabidopsis* protoplasts and in stable transgenic plants (Figure 5). The fusion protein was observed as dot-like structures and most of the signals were co-localized with fluorescence signals of RGP1 (Figure 5A–F), which is a trans-Golgi marker [43], suggesting that KOM was localized in the Golgi apparatus in plant cells.

### 2.7. KOM Was Required for the Accumulation of CalS5 during Pollen Development

In the Rhomboid family of proteins, several key amino acid residues are conserved throughout organisms [30]. Mutations at the Asn of TMD2, Ser of TMD4 and His of TMD6 abolish the protease activity of Rhomboids in vivo [44,45,46]. Of the three essential residues for the protease activity, the His of TMD6 was not conserved in KOM but was replaced by Asn (Figure 3D). In addition, two Ser residues were located in tandem in TMD4 in KOM. We investigated whether these residues are required for the biological function of KOM. First, we performed genetic complementation tests of the *kom-1* mutation using *KOM* genomic DNA (Appendix A). In this experiment, the Asn of TMD2, His of TMD6, the first Ser of TMD4, second Ser of TMD4 or both Sers of TMD4 were replaced by Ala (N141A, N241A, S187A, S188A and S187A/S188A, respectively). Unexpectedly, all mutated alleles were able to complement the *kom-1* mutation (Appendix A) and restored the aberrant pollen wall patterning of *kom-1* to the WT phenotype (Appendix A). This implied that KOM did not require proteolytic activity for its normal function. Consistent with this, in a standard Rhomboid activity assay, which we used previously to demonstrate the proteolytic activity of *Arabidopsis* Rhomboid AtRBL2 [31,46], KOM showed no proteolytic activity against classical Rhomboid substrates, including Spitz and Keren of Drosophila (Appendix A) and human TGFβ (data not shown). These findings suggest that the role of KOM in callose regulation did not depend on Rhomboid-like proteolytic activity.

The male sterility and callose wall abnormality observed in *kom-1* pollen grains closely resembled the phenotype of cals5, which is a mutant lacking the function of male-specific CasS5 [15,16]. To investigate a genetic interaction between *KOM* and *CalS5*, we crossed *kom-1* with a *cals5* mutant. Among the mutant alleles, *cals5-5*, which has a T-DNA insertion in the third intron, showed a weaker phenotype in callose accumulation and pollen wall patterning [16]. When we crossed *kom-1* with *cals5-5*, the *kom-1/cals5-5* double mutant showed disorganized pollen wall patterning, which was similar to those of the severe pollen phenotype of kom-1 (Figure 6A–C). This result suggested that *KOM* was epistatic to *CalS5* and raised the possibility that KOM may regulate the function of CalS5. It was reported that *CalS5* is expressed in the tapetum cells and meiosis to tetrad stages of microspore [17]. In our experimental condition, CalS5 signals were strongly detected in the microspores and weakly detected in the tapetum cells, both in WT and *kom-1* (Figure 6D–F), suggesting that *KOM* did not affect the *CalS5* expression. 

To determine whether KOM is required for the function of the CalS5 protein, we attempted to detect CalS5 protein in WT and mutant plants. We raised polyclonal antibodies against the N-terminal region of CalS5 (anti-CS5N) and its central loop region (anti-CS5L) for Western blot analysis. Both anti-CS5N and anti-CS5L detected a 220 kDa band in protein extracted from the WT membrane fraction, which corresponded to the molecular size of CalS5 protein. In contrast, no such band was detected in protein extracted from cals5, and a faint band was detected in those from *kom-1* (Figure 6G). Together, these findings indicated that KOM was required for CalS5 protein accumulation and had an essential function in the formation of the callose wall during pollen development.

## 3. Discussion

### 3.1. Pollen Wall Patterning Was Required for Pollen–Stigma Adhesion

Pollen–stigma adhesion is the first step of cell–cell interactions during fertilization in plants. Several factors, including pollen wall patterns, pollen coat components and stigmatic proteins, are involved in the pollen–stigma adhesion event [1,16,47,48]. Among these, pollen wall patterning is thought to play a major role in the initial pollen–stigma adhesion [1]. The *kom* mutation affects pollen sterility, but both female and male gametophytes can function normally. Data obtained from pollen–stigma adhesion assays showed that male sterility is caused by the weak adhesion of *kom-1* pollen grains to the stigma (Figure 1I). The pollen grains of *kom-1* contained apparently normal pollen coats but aberrant pollen wall patterning (Figure 1D,F). Therefore, the pollen wall patterning played a principal role in pollen–stigma adhesion and male fertility observed in the *kom* mutant.

### 3.2. KOM Was a Novel Regulator of Callose and Pollen Wall Formation

Several *Arabidopsis* mutants, such as *faceless pollen 1* (f*lp1*), *no exine formation* (*nef1)*, *defective in exine formation 1* (*dex1*) and *male-sterile 2* (*ms2*), were characterized as having defects in pollen wall development [2,3,4,6,7,8,49,50,51,52]. Abnormalities in these mutants are observed during sporopollenin deposition; therefore, these mutants are thought to be defective in exine components or sporopollenin aggregation patterns. 

Another class of pollen wall mutant comprises mutations that affect the synthesis of the callose wall, which provides a mold for pollen wall patterning. Mutations in the *CalS5* gene abolish the callose wall, causing male sterility [15,16,28]. In plants with *cals11/cals12* double mutations, the callose wall enclosing the pollen mother cells appears normal, but the wall separating tetrad microspores cannot be formed normally due to a lack of callose accumulation [12]. Transgenic plants expressing callase (β-1,3-glucanase) in the anther locule are unable to synthesize the callose wall during microsporogenesis, resulting in the production of pollen grains with aberrant wall patterning [23,24]. In the present study, *kom-1* lacked the callose necessary to enclose the pollen mother cells (Figure 2E,F). Because KOM shares no similarity with glucan synthases or glucanases, KOM represents a new class of proteins that modulate the activity of callose synthesis/degradation enzymes.

### 3.3. KOM May Represent a New Function for Rhomboid-like Protein 

It was reported that CDKG1, which encodes a cyclin-dependent kinase, is associated with a spliceosome to regulate *CalS*5 mRNA splicing and protein maturation [28]. We did not observe a splicing defect of *CalS*5 in *kom-1* mutant but observed a CalS5 protein accumulation defect (Figure 6G). Therefore, KOM may regulate CalS5 activity in a different way than CDKG1.

Since the role of Rhomboid proteins was first demonstrated in the Drosophila EGF signaling pathway [46,53], the most characterized function of Rhomboid proteases conserved through organisms, including bacteria, animals and plants, is the proteolytic cleavage of membrane proteins [30,31,54]. Later, this family of proteins was implicated in diverse cellular processes, including protein homeostasis, viral susceptibility, mitochondria membrane fusion and parasite–host interaction [34,39,40,55,56,57,58,59,60]. The genetic and biochemical data presented in this study suggest that KOM acts not as a Rhomboid protease but as a member of the growing sub-family of proteolytically inactive Rhomboid-like proteins (Appendix A) [61]. These pseudoproteases have a very wide range of biological functions, although current evidence suggests that a common mechanistic theme is that they interact specifically with TMDs, thereby regulating membrane proteins, often by affecting their stability. Although KOM is unlikely to have proteolytic activity itself, the loss of KOM showed the clear phenotype of CalS5 with callose accumulation. This is the first evidence that catalytically inactive Rhomboid-like protein has a function in plants.

Interaction with EGF ligands was characterized in catalytically inactive Rhomboid proteins [62]. Interestingly, the C-terminal lumenal domain of KOM is longer than those of other *Arabidopsis* Rhomboid-like proteins that show detectable serine protease activity against classical Rhomboid substrates from Drosophila [31]. Because CalS5 is a membrane protein and its accumulation is reduced in *kom-1*, KOM may act, for example, to stabilize CalS5 on a membranous component in meiocyte (Appendix A). Further characterization of KOM will provide new insights into the regulatory mechanisms of the Rhomboid family of proteins, callose accumulation and microspore development in plants.

## 4. Materials and Methods

### 4.1. Plant Material and Growth Conditions

Columbia-0 (Col-0) accession of *Arabidopsis thaliana* was used as wild-type plants (WT). The *kom-1* mutant was originally isolated from an EMS-mutagenized M2 population of Col-0. The mutant line used in this study was generated after two backcrosses to Col-0. T-DNA insertion mutants, namely, SALK_016980 (*kom-2*) (this study) and SALK_072226 (*cals5-5)* [16], were obtained from the *Arabidopsis* Biological Resource Center at Ohio State University (Columbus, OH, USA). The hemizygous *kom-1/kom-2* plant showed the same phenotypes as *kom-1* and *kom-2* homozygous plants. All the qualitative data presented in this manuscript are representative of the phenotypes of these two alleles.

Seeds were sown on the surface of vermiculite in small pots and incubated for four days at 4 °C. Plants were grown under continuous white light at 22–24 °C. 

### 4.2. Microscopy 

For transmission electron microscopy (TEM), flowers were fixed in a 50 mM sodium cacodylate buffer containing 4% paraformaldehyde (Sigma-Aldrich, Tokyo, Japan) and 1% glutaraldehyde (Sigma-Aldrich, Tokyo, Japan) for 12 h. Samples were kept under gentle vacuum conditions for the first hour at room temperature and were then maintained at 4 °C. After fixation, samples were washed with 50 mM sodium cacodylate buffer twice and post-fixed in 50 mM sodium cacodylate buffer containing 1% osmium tetroxide (Sigma-Aldrich, Tokyo, Japan) for three hours at 4 °C. After washing with 50 mM sodium cacodylate buffer twice and distilled water once, they were dehydrated with a graded ethanol series and kept in 100% ethanol overnight at 4 °C. They were subsequently treated with propylene oxide (Sigma-Aldrich, Tokyo, Japan) containing 5, 10, 15, 20, 30, 50 and 80% Spurr’s resin and then maintained in 100% Spurr’s resin (Funakoshi, Tokyo, Japan) overnight at 4 °C. Anthers were then dissected and embedded in Spurr’s resin. Ultra-thin (60–80 nm) sections were cut with a diamond knife, stained with uranyl acetate and lead citrate (Sigma-Aldrich, Tokyo, Japan), and observed with an SM-100SX transmission electron microscope (JEOL, Tokyo, Japan). For the pollen observation, flowers were fixed in a 9:1 ratio of ethanol to acetic acid overnight, treated with 90% and 70% ethanol for 20 min each, cleared in Hoyer’s solution (7.5 g gum Arabic (Sigma-Aldrich, Tokyo, Japan), 100 g chloral hydrate (Tokyo-Kasei, Tokyo, Japan), 5 mL glycerol (Sigma-Aldrich, Tokyo, Japan) in 30 mL water) and observed with Nomarski optics. For histological analyses, inflorescences were fixed in FAA solution (formaldehyde/acetic acid/alcohol) overnight, replaced with 50, 70, 80, 90, 99 and 100% ethanol, and embedded in Technovit 7100 resin (Kulzer, Heraeus, Germany). Sections that were 5 µm thick were double-stained with 0.1% Toluidine blue (Cosmo-Bio, Tokyo, Japan) and 1% aniline blue (Chroma, Münster, Germany) and observed under UV illumination.

### 4.3. Pollen–Stigma Adhesion Assay

Col-0 or *kom-1* stamens were immobilized on a table with double-sided tape. Col-0 pistils that had been emasculated to prevent self-pollination were carefully touched to the stamens. Pollinated pistils were vortexed in a buffer and the numbers of pollen grains detached into the buffer and remained on the stigma were counted 10 s or 1 h after pollination. Average values were obtained from five individual assays. 

### 4.4. Mapping and Cloning of the KOM Gene 

Approximately 4000 F2 plants from a cross between *kom-1* and L*er* were used for mapping the *KOM* locus. DNA markers used for positional cloning were based on an SSLP (simple sequence length polymorphism) between ecotypes Col-0 and L*er*. Information about T5M6-1, F28K19-1, F28K19-2 and F28K19-3 markers was obtained from the MONSANTO *Arabidopsis* Polymorphism and L*er* Sequence Collection (http://www.Arabidopsis.org/Cereon/index.jsp (accessed on 28 March 2022)). 

### 4.5. Complementation Test 

A 3.6 kb genomic fragment spanning the *KOM* gene, corresponding to region 24824–28559 of the BAC clone F28K19, was amplified using PCR (primer sequences 5′-GAGCAATGATTTTCTCCTTGAGAGATGC-3′ and 5′-GAGATACAACTCTTCGGAAAGG-3′) and cloned in pBluescript II SK (+) (Toyobo, Japan). This genomic fragment included 664 bp of the 5′ region, 1042 bp of the 3′ region and the whole ORF of the At1g77860 gene. The fragment was digested with *Kpn* I and cloned into binary vector pPZP211 [63] to generate pPZP211-KOM. *Agrobacterium tumefaciens* C58C1 containing pPZP211-KOM was used to transform *kom-1* plants using a vacuum infiltration procedure. Transgenic plants were selected on an agar medium containing 30 µg/mL kanamycin (Sigma-Aldrich, Tokyo, Japan) and 100 µg/mL carbenicillin (Sigma-Aldrich, Tokyo, Japan). Pollen phenotypes were examined for complementation. 

For the cite-directed mutagenesis of KOM genes, the 3.6 kb KOM genomic DNA was amplified via PCR (using ATGTCGAATTCATCTGCGAGTCTGAACTTC and CAACTGTCGACAAAACAGA-GTAGGTCTTCG) and cloned into pBluescript II SK (+) at *Eco*R I—*Sal* I site. This vector was used as a template of the PCR to generate site-directed mutation alleles of N141A (using TTTCATCTATTCATAGCTCTTGGGAGTTTG and TAATCCACTGTGCAGCCATGGAG), N241A (CTTTGCAGCTATTGGTGGTTTCATATCAGG and TTGTCTATGAAAGGG-AGAAAGCCTA), S187A (CATCAATCGCTTCTGGTGCTGC and GGATGTTCCGAACA-AACAACACAGC), S188A (CATCAATCTCTGCTGGTGCTGC and GGATGTTCCGAAC-AAACAACACAGC) and S187A/S188A (CATCAATCGCTGCTGGTGCTGC and GGATGTTCCGAACAAACAACACAGC). PCR products containing the mutated KOM genes were self-ligated and cloned into the binary vector pPZP211 at the *Eco*R I—*Sal* I site. The constructs were used to transform *kom-1* plants via the *Agrobacterium tumefaciens* C58C1 strain-mediated floral dip approach. Transgenic plants were selected on an agar medium containing 30 µg/mL kanamycin and 100 µg/mL carbenicillin. Pollen phenotypes were examined for complementation in at least three independent lines. 

### 4.6. Peptide Sequence Analysis

Alignments were performed using the ClustalW algorithm [64]. Transmembrane domains were predicted according to the TMHMM (http://www.cbs.dtu.dk/services/TMHMM/ (accessed on 6 July 2007)) and TMPred (http://www.ch.embnet.org/software/TMPRED_form.html (accessed on 6 July 2007)) algorithms.

### 4.7. cDNA Cloning and RT-PCR

Total RNA isolation and first-strand cDNAs synthesis were done as described previously [31]. *KOM* cDNA was obtained by both 5’ and 3’ RACE using the SMART RACE cDNA Amplification Kit (TaKaRa, Tokyo, Japan). The accession number of *KOM* cDNA is AB161192. 

For RT-PCR, cDNAs were synthesized from 0.9 µg of total RNA with the SuperScript II First-Strand Synthesis System (Thermo Fisher Japan, Tokyo, Japan). The tissues used were the aerial parts from seedlings 10 days after germination, roots from seedlings 10 days after germination, rosette leaves, cauline leaves, stems, siliques and inflorescences. KOM-specific primers (CATTATGGGTATGCTGTTTGC and CTTGTGAGATATTGGTGAAAGG) and actin (ACT8)-specific primers (An et al., 1996) were used to amplify *KOM* and *ACT8* cDNAs using PCR for 35 cycles.

### 4.8. In Situ RNA Hybridization

Expression of KOM and CalS5 in developing flowers was detected using in situ mRNA hybridization as described previously [65,66]. Inflorescences were fixed with 4% paraformaldehyde in PBS. Paraffin sections (8 µm thick) were hybridized with digoxygenin-labeled probes. Antisense and sense probes of *KOM* and *CalS5* were prepared using fragments of their cDNAs that had been amplified using PCR and cloned into pBluescript II SK (+) at the Sal I site. The probe sequences used for hybridization corresponded to KOM cDNA between 59 and 569 bp and CalS5 cDNA between 4864 and 5414 bp. 

### 4.9. Subcellular Localization Analysis 

To make GFP-tagged constructs, a PCR fragment of G3GFP was used to replace the *GUS* gene of a binary vector pBI121 to generate p35SG3GFP. *KOM* cDNA was inserted between the CaMV 35S promoter and G3GFP at an *Xba* I site to generate p35S-KOM-GFP. For transient expression in *Arabidopsis* protoplasts, the *Hin*d III—*Eco*R I fragment of p35S-KOM-GFP, which contained the CaMV 35S promoter, KOM-GFP and NOS-terminator, was cloned into pBluescript II SK (+). Transformation of *Arabidopsis* Col-0 suspension culture cells [67] was performed as described previously [68]. Transformed protoplasts were incubated under gentle agitation at 23 °C for at least 8 h in the dark. Immunofluorescent staining was performed as described [69,70]. A 500-fold dilution of anti-RGP1 antibodies [43] was used to stain the RGP1 protein, which is a Golgi marker. Alexa-Fluor^TM^-546-conjugated goat antibodies against rabbit IgG (100-fold dilution; Molecular Probes, Eugene, OR, USA) were used as secondary antibodies. Subcellular localization of proteins was observed with a confocal laser microscope system (LSM510, ZEISS, Jena, Germany) with the 488 nm line of an Ar/Kr laser for GFP and the 543 nm line of a He/Ne laser for Alexa Fluor^TM^ 546. The transient expression assay was repeated more than five times to confirm the protein localization.

### 4.10. Western Blotting

Inflorescences (1.0 g) from Col-0, *kom-1* and *cals5-5* were ground to a fine powder in liquid nitrogen. Powders of the samples were homogenized in extraction buffer with proteinase inhibitor (20 mM Tris-HCl, pH 7.5, 50 mM NaCl, 1 mM EDTA, 1 mM 2-mercaptoethanol, 0.5 mM phenylmethylsulfonyl fluoride (Signa-Aldrich, St. Luis, MO, USA)). The homogenates were centrifuged at 100,000× *g* for 60 min to separate the soluble and total membrane fractions. The same amount of proteins in each sample was dissolved with an SDS sample loading buffer and resolved on SDS-PAGE. Western blot analysis was performed using an anti-CalS5-N antibody (corresponding to CalS5 amino acid position 80–180) and anti-CalS5-L (corresponding to amino acid positions 898–1025).

## 5. Conclusions

Here, we present genetic and molecular data showing that mutation in the *KOMPEITO* gene caused pollen wall morphology and pollen–stigma adhesion. KOM falls within a sub-class of Rhomboid-like proteins that have lost protease activity. *KOM* was predominantly expressed in the meiocyte and loss of *KOM* expression affected CalS5 protein accumulation. Taken together, *KOMPEITO* was required for proper callose wall formation via affecting CalS5 protein function in the meiocyte, as well as for pollen wall formation and plant fertilization.

## Figures and Tables

**Figure 1 ijms-23-05959-f001:**
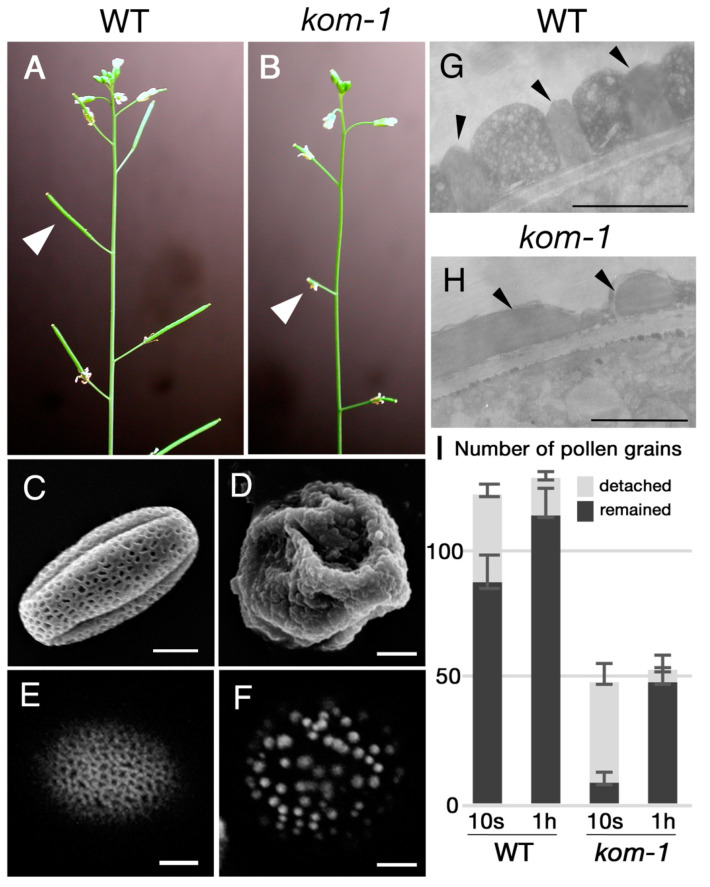
Phenotypes of *kom-1* inflorescence and mature pollen grains. (**A**) Wild-type (WT) plant with long seed pots (arrowhead). (**B**) *kom-1* plant with shorter pots (arrowhead). (**C**,**D**) SEM images of mature pollen grains of the WT (**C**) and *kom-1* (**D**). (**E**,**F**) Confocal microscope images of Auramine-O-stained mature pollen grains of the WT (**E**) and *kom-1* (**F**). Note that the mesh-like patterns of the WT (**E**) and the dot-like structures of *kom-1* pollen surface (**F**) were both made of sporopollenin, as they were stained by Auramine-O. (**G**,**H**) TEM images showing pollen wall sections of the WT (**G**) and *kom-1* (**H**). The exine structure of mature *kom-1* pollen grains was altered (arrowheads). (**I**) Pollen–stigma adhesion assay. Numbers of WT and *kom-1* pollen grains that were attached to a WT pistil and detached from the pistil after washing (10 s or 1 h) were counted. The values represent the average of 5 independent trials. Bars: (**C**–**F**), 5 µm; (**G**,**H**), 1 µm.

**Figure 2 ijms-23-05959-f002:**
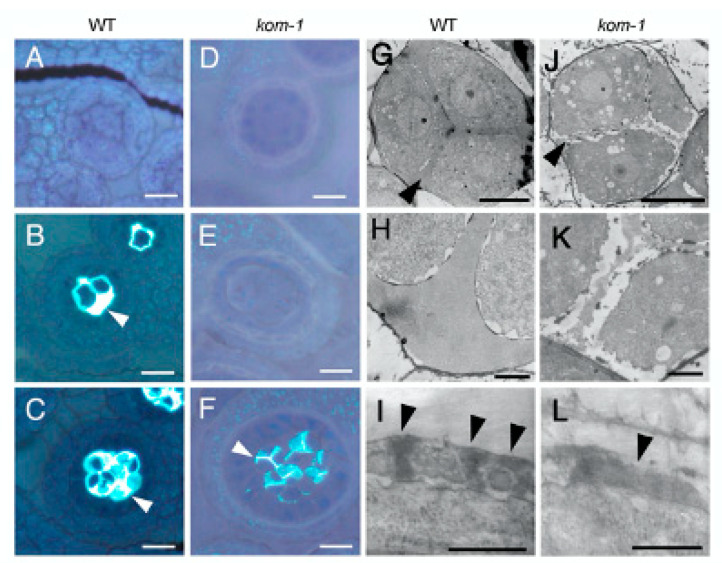
Phenotypes of *kom-1* pollen mother cells and microspores. (**A**–**F**) Callose accumulation of the WT (**A**–**C**) and *kom-1* (**D**–**F**). Sections of anthers were stained with toluidine blue-O and aniline blue to visualize cells and callose, respectively. (**A**,**D**) Anthers before meiosis. (**B**,**E**) Pollen mother cells undergoing meiosis were surrounded by the callose wall (arrowhead) in the WT (**B**), but not in *kom-1* (**E**). (**C**,**F**) Tetrad microspores were surrounded by the callose wall (arrowhead) in the WT (**C**). In *kom-1*, callose walls were present only between microspores (arrowhead) but were absent in the outer callose wall surrounding a tetrad (**F**). (**G**–**L**) TEM images of tetrad microspores of the WT (**G**–**I**) and *kom-1* (**J**–**L**). Callose walls (arrowhead) surrounding the microspores were thick in the WT (**G**) but thin in *kom-1* (**J**). (**H**,**K**) Higher magnification of (**G**,**J**). Sporepollenin was aggregated to form pre-exine (arrowheads) in WT (**I**) but was deposited in irregular patterns in *kom-1* (**L**). Bars: (**A**–**F**), 20 µm; (**G**,**H**,**J**,**K**), 5 µm; (**I**,**L**), 0.5 µm.

**Figure 3 ijms-23-05959-f003:**
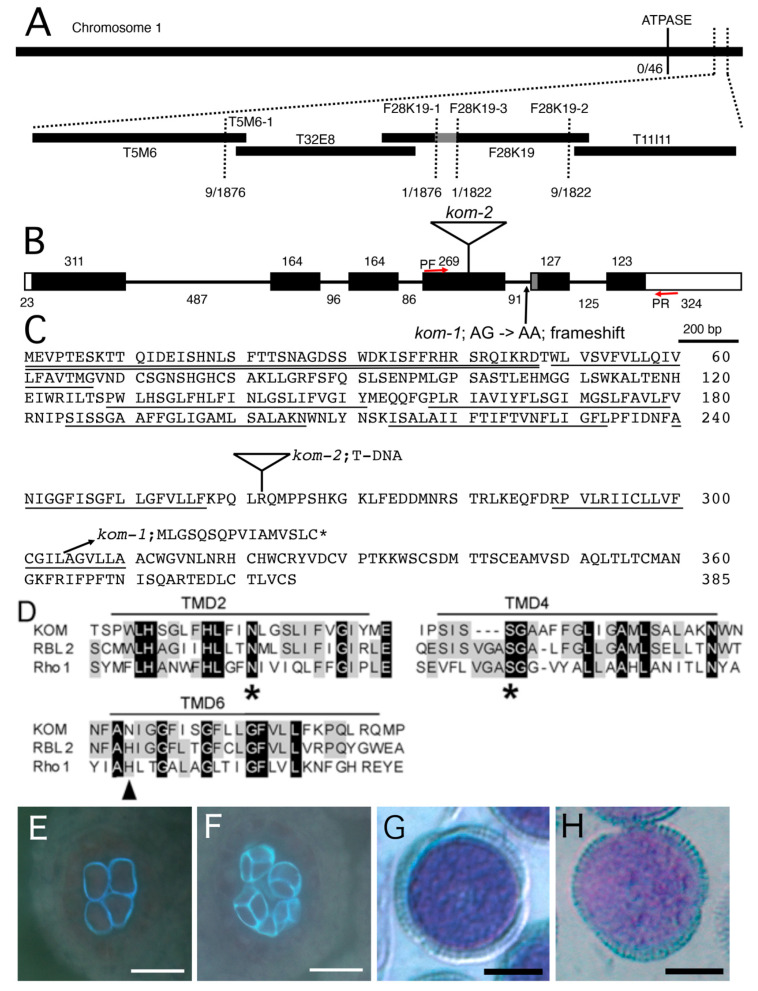
Cloning and complementation of the *kom-1* mutation. (**A**) Map position of the *kom-1* mutation. The *kom-1* locus was mapped to the lower arm of chromosome 1 between F28K19-1 and F28K19-3 markers on BAC F28K19. The 12.5 kb genomic DNA between the two markers contained only one nucleotide substitution (**A**–**G**) in the At1g77860 locus, which is one of the four open-reading frames (ORFs) in this region. (**B**) DNA structure of *KOM* gene and mutation point in the *kom-1* and *kom-2* alleles. White boxes and black boxes represent untranslated regions and exons, respectively. The G-to-A substitution in *kom-1* occurred in the splicing acceptor (**A**,**G**) of the fourth intron of the *KOM* gene (At1g77860) that consisted of six exons. An alternative splicing acceptor that was 23 bp downstream of the original site (the region drawn in gray) was used instead in *kom-1*, causing a frameshift that truncated the C–terminal 70 residues of KOM peptide. The *kom-2* allele (SALK_016980) contained a T-DNA insertion in the fourth exon. Red arrows indicate the forward (PF) and reverse (PR) primers for RT-PCR. (**C**) KOM protein sequence (GenBank accession number for *KOM* cDNA, AB161192) and predicted *kom-1*protein sequence. A white triangle represents a T-DNA insertion site in *kom-2*. Double underline and underlines represent the signal anchor region and transmembrane domains (TMD), respectively. (**D**) Alignment of predicted KOM amino acid sequence with Drosophila Rhomboid-1 (DmRho-1, accession No. NM_079159) and AtRBL2 (AB195671). Only segments of three transmembrane domains (TMD2, TMD4 and TMD6) are shown. The amino acid residues conserved in all of the three sequences are highlighted in black, and those conserved in at least two of the three sequences are highlighted in gray. Stars (*) indicate the highly conserved Asn in TMD2 and Ser in TMD4. Arrowhead indicates the His in TMD6 that is conserved in the Rhomboid family but is replaced by Asn in KOM. (**E**–**H**) Complementation test. The *kom-1* plants were transformed with a 3.6 kb genomic DNA fragment containing the *KOM* gene. Phenotypes in callose accumulation and pollen surface patterning were scored for a complementation test. Transgenic plants formed a normal callose wall in pollen mother cells undergoing meiosis (**E**) and in a tetrad (**F**), as well as normal surface patterning of mature pollen grains (**G**) identical to that of the WT (**H**). Bars: (**E**,**F**), 20 µm; (**G**,**H**), 10 µm.

**Figure 4 ijms-23-05959-f004:**
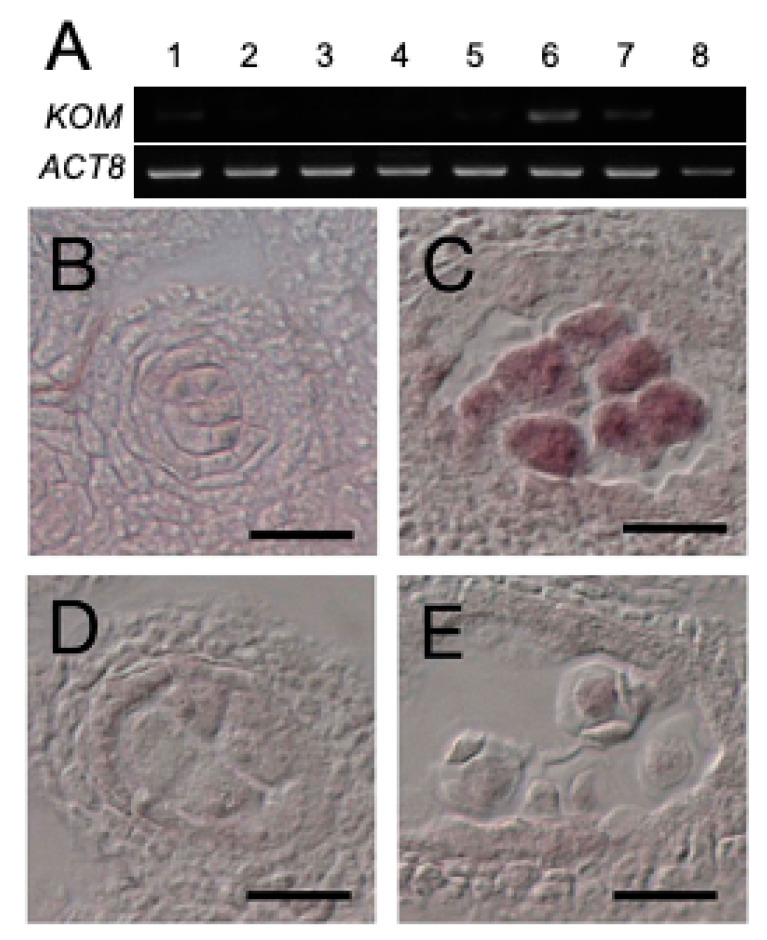
*KOM* gene expression. (**A**) Analysis of *KOM* expression in different plant tissues using RT-PCR. Samples used were total RNA from aerial parts of 10-day-old seedlings, roots, rosette leaves, cauline leaves, stems, siliques, inflorescences and genomic DNA (lanes 1 to 8, respectively). *ACT8* was used to normalize the RT-PCR. (**B**–**E**) Analysis of the *KOM* expression in the anther using in situ hybridization. *KOM* expression was detected in the pollen mother cells (PMC) undergoing meiosis (**C**) but not in the PMC before meiosis (**B**) nor in the microspore tetrads (**D**). No signal was detected in the meiotic PMC when the sense mRNA of *KOM* was used as the negative control (**E**). Bars: 20 µm.

**Figure 5 ijms-23-05959-f005:**
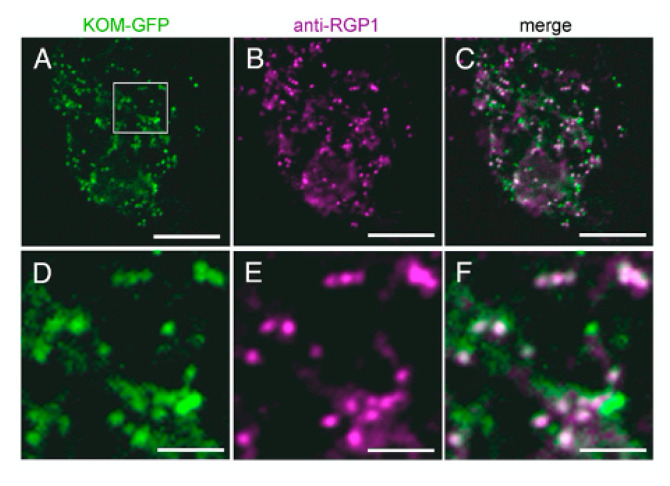
Subcellular localization of KOM. (**A**–**F**) The localization of KOM was observed in protoplasts transiently expressing KOM-GFP. Images of KOM-GFP green fluorescence (**A**) and RGP1 purple signals (**B**) were merged to generate the overlay image (**C**). Higher magnification of the boxed area in (**A**–**C**) was presented in (**D**–**F**), respectively. Bars: (**A**–**C**), 20 µm; (**D**–**F**), 50 µm.

**Figure 6 ijms-23-05959-f006:**
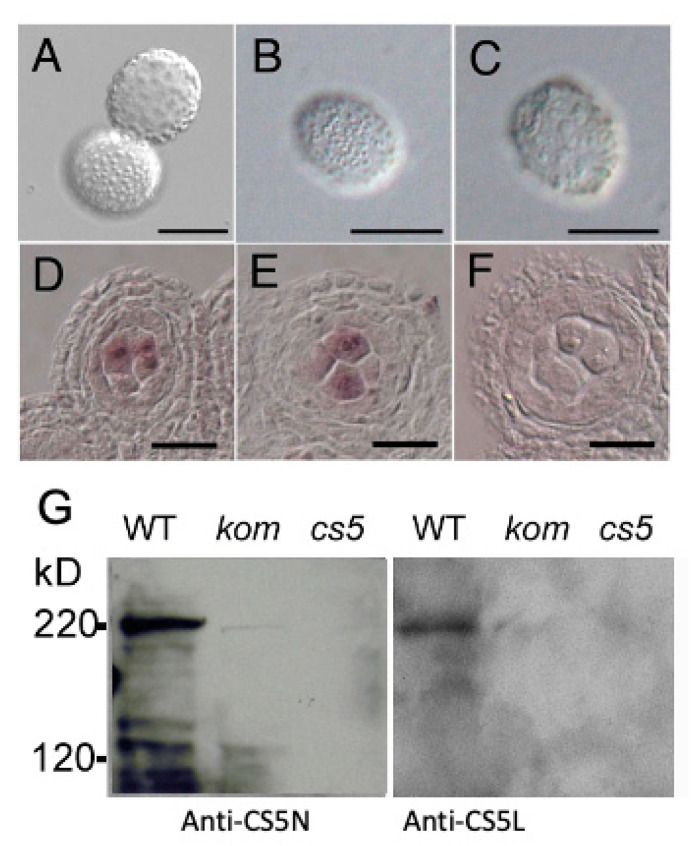
KOM affected CalS5 protein accumulation but not *CalS5* mRNA expression. (**A**–**C**) The defect in pollen wall patterning was comparable in *kom-1* (**A**), *cals5-5* (**B**) and *kom-1/cals5-5* double mutants (**C**). (**D**–**F**) Analysis of *CalS5* expression in anthers using in situ mRNA hybridization. *CalS5* mRNA was expressed in pollen mother cells undergoing meiosis in the WT (**D**), as well as in *kom-1* (**E**). Sense mRNA of *CalS5* was used as the negative control (**F**). (**G**) Western blot analysis of proteins from floral buds using polyclonal antibodies against CalS5-N and CalS5-L. CalS5 was detected as a 220 kDa protein in the WT flower buds, but it was less prevalent in *kom-1* (kom) and absent in the *cals5-5* (cs5) null allele. Bars: 20 µm.

## Data Availability

The data presented in this study are available on request from the corresponding author.

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
