# Peer review of "KOMPEITO, an Atypical Arabidopsis Rhomboid-Related Gene, Is Required for Callose Accumulation and Pollen Wall Development"

_ijms, 2022, doi:10.3390/ijms23115959_

Round 1

Reviewer 1 Report

This manuscript reported the isolation of a pollen wall defective mutant and its importance to form normal pollen callose walls. I believe that Experimental approaches and interpretation are scientifically reasonable. Thus, I strongly recommend that this manuscript must be accepted as it is.  

Author Response

Reviewer 1 Comments for the Author

This manuscript reported the isolation of a pollen wall defective mutant and its importance to form normal pollen callose walls. I believe that Experimental approaches and interpretation are scientifically reasonable. Thus, I strongly recommend that this manuscript must be accepted as it is.  

We are very glad to hear your positive comment for our manuscript. Although you accepted the current version of the manuscript, we modified the manuscript based on the suggestions from other reviewers.

Reviewer 2 Report

The article for review: "KOMPEITO, an atypical Arabidopsis Rhomboid-related gene, is required for callose accumulation and pollen wall development," is interesting and written in a readable style.
In this study, a kompeito-1 mutant was isolated, defective in pollen wall modeling, and pollen adhesion-a stigma in Arabidopsis thaliana. Then the authors perform genetic and molecular analyzes on a new member of the Rhomboid protein family to elucidate its role in callose accumulation and pollen wall development.
All the results presented by the authors show that KOM is required for the accumulation of CalS5 protein, which leads to the regulation of meiocyte-specific accumulation of calluses and pollen wall formation.
The manuscript covers all the requirements of the IJMS journal. The results can be used further to analyze plants' callose wall formation and pollen development.

It is necessary to improve the quality of figures: 1I and 3A, B, C, D!
References have duplicate numbers!

Author Response

Reviewer 2 Comments for the Author

The article for review: "KOMPEITO, an atypical Arabidopsis Rhomboid-related gene, is required for callose accumulation and pollen wall development," is interesting and written in a readable style.
In this study, a kompeito-1 mutant was isolated, defective in pollen wall modeling, and pollen adhesion-a stigma in Arabidopsis thaliana. Then the authors perform genetic and molecular analyzes on a new member of the Rhomboid protein family to elucidate its role in callose accumulation and pollen wall development.
All the results presented by the authors show that KOM is required for the accumulation of CalS5 protein, which leads to the regulation of meiocyte-specific accumulation of calluses and pollen wall formation.
The manuscript covers all the requirements of the IJMS journal. The results can be used further to analyze plants' callose wall formation and pollen development.

It is necessary to improve the quality of figures: 1I and 3A, B, C, D!
References have duplicate numbers!

We are very glad to hear your positive comment for our manuscript. We have uploaded figures with better quality. We also carefully checked References and numbers.

Reviewer 3 Report

In this manuscript, Kanaoka et al. reported an Arabidopsis mutant, kom-1, with defection in pollen wall patterning and pollen-stigma adhesion. Through mapping cloning and T-DNA insertion mutant analysis, the authors proved that the KOM gene encodes a member of Rhomboid serine protease proteins. Notably, mutations in a callose synthase gene, CalS5, also showed the lack of callose accumulation phenotype. Further studies showed that KOM is required for CalS5 protein accumulation, instead of transcription. This finding reveals a new genetic factor in regulating callose accumulation and pollen wall development. Here, I have a few concerns about this manuscript, which is listed in detail as follows. 

  1. In Results 2.1, to support "kom-1 pollen tubes germinated and grew normally", in vitro pollen germination assay would be a good option to compare mutant and WT pollen germination ratio.
  2. In Results 2.1, the authors concluded that the female gametophytes of kom-1 mutant developed normally by providing Supplemental Figure S1A,B. However, the images in Figure S1A,B are not clear enough to show that it is normal, in terms of synergid cells, egg cell, central cell nuclei, and antipodal cells. I would suggest pollinating kom-1 mutant with WT pollens. The rescue of seed abortion in kom-1 would be a strong evidence for normal female gametophyte.
  3. In Results 2.2, the author proposed a question "whether the KOM gene is expressed during pistil formation". No experiment is carried out here to directly answer this question.
  4. About the complementation test of the kom-1 mutant, the length of the promoter and terminator used for complementation should be provided in either Results or Materials and Methods.
  5. In Results 2.4, to prove that "KOM is a unique member of the Rhomboid family of proteins", phylogenetic analysis with full length of more Rhomboid family proteins should be provided.
  6. In Figure 4A, it is hardly to see the bands in lanes 2 and 3. qPCR is a much better way to present its expression pattern. In addition, it would be better to show the position of KOM-specific primers on the gene structure.
  7. In Figure 5, I didnot see Fig 5G-H as the author mentioned in the figure legend.
  8. In Figure 6G, a reference control should be provided to prove the same loading amount.

Author Response

Reviewer 3 Comments for the Author

In this manuscript, Kanaoka et al. reported an Arabidopsis mutant, kom-1, with defection in pollen wall patterning and pollen-stigma adhesion. Through mapping cloning and T-DNA insertion mutant analysis, the authors proved that the KOM gene encodes a member of Rhomboid serine protease proteins. Notably, mutations in a callose synthase gene, CalS5, also showed the lack of callose accumulation phenotype. Further studies showed that KOM is required for CalS5 protein accumulation, instead of transcription. This finding reveals a new genetic factor in regulating callose accumulation and pollen wall development. Here, I have a few concerns about this manuscript, which is listed in detail as follows.

We appreciate all of your suggestions. Below are our responses to your comments.

  1. In Results 2.1, to support "kom-1 pollen tubes germinated and grew normally", in vitro pollen germination assay would be a good option to compare mutant and WT pollen germination ratio.

We tried in vitro pollen tube germination assay. However, due to technical difficulty, the germination rate was not stable even in wildtype (Col) pollen grains; the germination rate is often influenced by local humidity of the germination chamber, pollen density on the medium, and so on. Instead, we did hand-pollination assay to show viability of kom-1 pollen grains. When we hand-pollinated kom-1 pollen grains to Col pistils, all the ovules were fertilized and the silique was fulfilled with seeds, which is similar to self-pollination of Col pollen grains. This result suggested that kom-1 pollen grains are fertile.  We added pictures to Supplementary Figure S1G, and main text was modified accordingly.

  1. In Results 2.1, the authors concluded that the female gametophytes of kom-1 mutant developed normally by providing Supplemental Figure S1A,B. However, the images in Figure S1A,B are not clear enough to show that it is normal, in terms of synergid cells, egg cell, central cell nuclei, and antipodal cells. I would suggest pollinating kom-1 mutant with WT pollens. The rescue of seed abortion in kom-1 would be a strong evidence for normal female gametophyte.

We pollinated wildtype (Col) pollen grains to kom-1 pistils and, like wildtype self-pollination, the mutant pistils were fulfilled with seeds. This suggests that kom-1 female gametophyte has no defect in fertilization and seed formation. We added pictures to Supplementary Figure S1G, and main text was modified accordingly.

  1. In Results 2.2, the author proposed a question "whether the KOM gene is expressed during pistil formation". No experiment is carried out here to directly answer this question.

We did in situ mRNA hybridization to show spacial expression of KOM, and KOM was expressed in the megaspore mother cells during meiosis (Fig S2G-I). This result was described in Results 2.5. we deleted the question whether the KOM gene is expressed during pistil formation" from Results 2.1.

  1. About the complementation test of the kom-1 mutant, the length of the promoter and terminator used for complementation should be provided in either Results or Materials and Methods.

The promoter and terminator used for complementation test were 664bp and 1042bp in length, respectively. We added this information in Materials and Methods 4.5.

  1. In Results 2.4, to prove that "KOM is a unique member of the Rhomboid family of proteins", phylogenetic analysis with full length of more Rhomboid family proteins should be provided.

We made a phylogenetic tree using KOM and seven Rhomboid-related proteins in Arabidopsis thaliana. Drosophila Rhombid-1 protein was used as an outgroup. As a result, KOM located most outside in Arabidopsis Rhomboid-related proteins, which suggests KOM is a unique member of the Rhomboid family protein. The phylogenetic tree was presented as Supplementary Figure S3, and Figure S3 in original manuscript was renamed as Figure S5.

  1. In Figure 4A, it is hardly to see the bands in lanes 2 and 3. qPCR is a much better way to present its expression pattern. In addition, it would be better to show the position of KOM-specific primers on the gene structure.

We are sorry to say we cannot do qRT-PCR in our laboratory now. Instead, we decided to present publicly-available gene expression data.  Based on data from eFP blowser (Fig S4), KOM is expressed in limited tissues such as rosette leaves, shoot apex, flower buds and open flowers, but the expression is low. The tissue which KOM has highest expression is flower15-19, which corresponds to meiosis stage in male gametogenesis. This RNAseq data confirms our RT-PCR data. We also added primer information in Fig 3.

  1. In Figure 5, I didnot see Fig 5G-H as the author mentioned in the figure legend.

We apologize our mistake to present old version of figure legend. We delete these sentences.

  1. In Figure 6G, a reference control should be provided to prove the same loading amount.

We have loaded the same amounts of proteins (1 µg) for each lane. We added this information to Materials and Methods 4.10.

Reviewer 4 Report

KOMPEITO, an atypical Arabidopsis Rhomboid-related gene, is required for callose accumulation and pollen wall development

Manuscript ID: ijms-1684704

Here are my comments (but not limited to):

The numbering of lines is not active, which makes it difficult to assign comments to the target part of this manuscript. For this reason, Most of the comments will be added to the PDF version of the manuscript and highlighted yellow.

  1. The authors should carefully check missing connecting words in the abstract and the rest of manuscript.
  2. In the abstract, specify the WT background genotype.
  3. In the last paragraph of the introduction section, the authors should first make a statement indicating the hypothesis of the study…such as characterizing the target genes under specific conditions prior to exposing what they did.
  4. 1.: this subtitle is too long. Consider revising to a more convenient one.
  5. In the last paragraph under 2.1., the authors should support the claim with genotyping data of tested genotypes proving their homozygosity.
  6. The authors should consider proposition subtitles in the discussion section that convey a clear message highlighting their findings. I found the current title generic with less insights. The authors should try to discuss the significance of their findings in a broad context, not just reflecting what other studies reported on the subject. This applies to 3.2, 3.3,…
  7. The authors should consider propose a signaling model summarizing their findings
  8. Regarding the complexity of the study, a clear conclusion should be drawn, and a take-home message delivered for the readers.
  9. Find more comments in the PDF version of the manuscript.

Author Response

Reviewer 4 Comments for the Author

The numbering of lines is not active, which makes it difficult to assign comments to the target part of this manuscript. For this reason, Most of the comments will be added to the PDF version of the manuscript and highlighted yellow.

We apologize the inconvenience for reviewing the manuscript.

  1. The authors should carefully check missing connecting words in the abstract and the rest of manuscript.

We carefully checked the whole manuscript and corrected the text. All the revisions to the manuscript are marked up using the “Track Changes” function of MS Word.

  1. In the abstract, specify the WT background genotype.

We used Col-0 as wildtype. We added this information in the abstract.

  1. In the last paragraph of the introduction section, the authors should first make a statement indicating the hypothesis of the study…such as characterizing the target genes under specific conditions prior to exposing what they did.

We added a brief summary of this manuscript (aim, hypothesis of this study, and so on).  

  1. : this subtitle is too long. Consider revising to a more convenient one.

We assume the subtitle is Results 2.1. We modified and shortened it to “Isolation of kompeito-1 (kom-1), which is defective in pollen wall and pollen-stigma adhesion”.

  1. In the last paragraph under 2.1., the authors should support the claim with genotyping data of tested genotypes proving their homozygosity.

When kom-1 was crossed to WT, all the F1 plants had wildtype phenotype. In F2 generation obtained from self-crossing of F1 plants, 24 out of 97 plants showed mutant phenotype, which suggests kom is a sporophytic recessive mutant. We added this to the last paragraph of Results 2.1.

  1. The authors should consider proposition subtitles in the discussion section that convey a clear message highlighting their findings. I found the current title generic with less insights. The authors should try to discuss the significance of their findings in a broad context, not just reflecting what other studies reported on the subject. This applies to 3.2, 3.3,…

We modified the subtitles of Discussion based on our findings.

  1. The authors should consider propose a signaling model summarizing their findings

We proposed a model for KOM function in Supplementary Figure S6.

  1. Regarding the complexity of the study, a clear conclusion should be drawn, and a take-home message delivered for the readers.

We added a paragraph for conclusion at the end of Discussion section.

  1. Find more comments in the PDF version of the manuscript.

We also appreciate your comments in the pdf file. We modified the manuscript according to your suggestion. As you mentioned, Discussion section 3.3. is repetitive and does not contain important message, so we decided to delete the section.

Round 2

Reviewer 3 Report

No further comments.

Author Response

We checked our manuscript and corrected some errors.

Reviewer 4 Report

The authors have taken into consideration most of the comments raised in the previous version. The manuscript has been improved significantly. However, the authors should put the conclusion statement after the materials and methods, not before. In the conclusion, a take-home message should be clearly delivered for the readers.

Author Response

We appreciate very much for your effort to check our revised manuscript. We prepared “Conclusions” section after “Materials and Methods” section.